# The effectiveness of putative wearable repellent technologies to protect against mosquito biting and *Aedes*-borne diseases, and their economic impact

**Robert T. Jones** [1]*, **Scott J. Tytheridge**[1,2], **Carolin Vegvari**[3,4], **Hannah R. Meredith**[5], **Elizabeth A. Pretorius**[1], **Thomas H. Ant**[6], **James G. Logan**[1]

**1** Department of Disease Control, London School of Hygiene & Tropical Medicine, London, United Kingdom, **2** The Ear Institute, University College London, London, United Kingdom, **3** Faculty of Medicine, School of Public Health, Imperial College, London, United Kingdom, **4** Oriole Global Health, London, United Kingdom, **5** Duke Global Health Institute, Durham, North Carolina, United States of America, **6** Centre for Virus Research, University of Glasgow, Glasgow, United Kingdom

* robert.jones@lshtm.ac.uk

**Data Availability Statement:** Data generated in this study is made available without restriction

## Abstract

Viruses transmitted by mosquitoes threaten the health of millions of people worldwide. There is an urgent need for new tools for personal protection to ensure that vulnerable individuals are protected from infectious bites when outdoors. Here, we test the efficacy of wash-in and spray-on repellents against *Aedes aegypti*. When applied as a treatment on clothing as well as skin, the novel repellent compound delta-undecalactone provided up to 100% protection initially, and over 50% bite prevention for more than 7 hours. Mathematical modelling indicated that if such a repellent, with 100% initial efficacy, were to be applied twice daily by 80% of the population, more than 30% of Zika virus infections could be averted in an outbreak scenario with a basic reproduction number $R_0 = 2.2$. In a less severe outbreak ($R_0 = 1.6$), the same repellent regimen could avert 96% of infections. If there was much lower uptake, with only 40% of people using the repellent twice per day, just 4% of Zika cases would be averted (outbreak $R_0 = 2.2$). Similar results were found in other scenarios tested for dengue and chikungunya outbreaks. Our model can be extrapolated to other repellents and guide future product development, and provides support to the concept that effective repellents that are used regularly and appropriately could be cost-effective interventions to prevent ill health from arboviral diseases.

## Author summary

Mosquito-borne diseases threaten millions worldwide, necessitating effective personal protection tools. Our study assessed wash-in and spray-on repellents against *Aedes aegypti* mosquitoes, the vectors of Zika and other arboviruses. When applied as a treatment on clothing and skin, the novel repellent compound delta-undecalactone provided over 50% bite prevention for more than 7 hours. Mathematical modelling indicated that if such a

through LSHTM Data Compass (https://doi.org/10.17037/DATA.00004432).

**Funding:** This work was supported by the project ZikaPLAN, funded by the European Union's Horizon 2020 research and innovation programme under Grant Agreement No. 734584 (JGL). The funders did not have a role in study design, data collection and analysis, decision to publish, or preparation of the manuscript.

**Competing interests:** The authors have declared that no competing interests exist.

repellent, with 100% initial efficacy, were to be applied twice daily by 80% of the population, it could mitigate more than 30% of Zika virus infections in a severe outbreak. In a less severe outbreak scenario, it could avert 96% of infections. Similar results were found for dengue and chikungunya outbreaks. Our findings underscore mosquito repellents as highly cost-effective interventions in preventing mosquito-borne illnesses, and the model we present can be used to guide the future development of repellent-based personal protection tools.

## Introduction

Vector-borne diseases account for a significant amount of the global infectious disease burden. Viruses transmitted by *Aedes* mosquitoes have become of increasing concern, with both an expansion of the geographic range and frequency of arboviral epidemics being observed in recent decades [1,2]. Close to three million suspected and confirmed cases of dengue virus (DNV) infection were reported in the Americas in 2022, and outbreaks of significant magnitude have been reported in 2023 [3]. Autochthonous cases are also now reported from southern Europe [4]. Chikungunya virus (CHKV) has additionally caused millions of cases of acute febrile disease since a new epidemic strain emerged in 2004 [5], and spread of the Zika virus (ZKV) was recognised as a Public Health Emergency of International Concern for the cluster of microcephaly cases and other neurological disorders associated with the virus that were described in 2015–2016 [6]. Finally, the yellow fever virus causes the most severe mosquito-borne infection in the tropics and is responsible for an estimated 29,000–60,000 deaths in Africa and South America each year [7].

Yellow fever is prevented by an extremely effective vaccine [8], but such effective vaccines are not widely available for other *Aedes*-borne viruses. For these, and where vaccine coverage is low, the risk of transmission in urban areas can be reduced by eliminating potential mosquito larval development sites, and through personal preventive measures [9]. Bed nets are not considered an effective tool because *Aedes* mosquitoes are day-biting and most active during daylight hours, and insecticide resistance has evolved in many Aedes populations worldwide, compromising the success of control interventions [10].

Personal protection tools (PPT) typically act by preventing contact between the body and the insect vector, and range from mosquito nets and vaporizers to repellents and protective clothing [11]. Wearable PPTs are particularly attractive because they offer the possibility of protecting individuals during the day, when users are at work or school, and could easily be integrated into everyday routines. Some vulnerable groups, including military personnel and agricultural workers, use insecticide-treated clothing as a means of protection from biting insects [12,13], and there are opportunities to provide protection to other communities by making suitable products more widely available and appealing.

Following the Zika epidemic in the Americas, we conducted focus group discussions to investigate the preferences of women of child-bearing age with regards to wearable PPT [14,15]. The focus group studies confirmed an interest in repellent products that can be applied at home to the clothing of the user's choice. Participants in the groups indicated that a strong, chemical-like smell was undesirable, and that 'natural' odours, including citronellol and its derivative citronella, have a pleasant fragrance that would be accepted by most users. Indeed, a perceived natural product with lower efficacy was considered more desirable than a synthetic product with higher efficacy [14].

Here, we have investigated the technical feasibility of developing both wash-in repellent laundry additives and post-laundry textile sprays for application to clothing. Further, we have

used mathematical modelling to estimate the number of ZKV, DNV, and CHKV infections and symptomatic cases that can theoretically be avoided by applying a mosquito repellent to skin or clothes with different assumptions on efficacy, half-life, coverage, and application frequency.

## Methods

### Mosquitoes

*Aedes aegypti* (pyrethroid susceptible strain) were obtained from a reference strain (originally from West Africa, colonised in 1926 with field additions in 1976) held at LSHTM, UK. All mosquitoes were reared and housed under optimal environmental conditions of 25°C ± 2°C and 80% RH with a 12: 12-hour photoperiod. All mosquitoes used in experiments were nulliparous females fed on 10% glucose solution.

### Participants

Arm-in-case tests were performed with three consented participants (author RTJ: demonstration of probing on covered skin, protection of covered and uncovered skin, and comparison of dUDL with PMD, authors EAP and TA: testing of repellents added to clothing in fabric softeners, selection of active compounds for spray treatments, and longevity of protection). The second arm of each participant was used as a control, with tests being repeated if the biting pressure was insufficient. No ethics committee approval was required for development testing of this nature that did not involve the recruitment of external participants.

### Demonstration of probing on covered skin

To demonstrate that mosquitoes might probe the skin of someone who is wearing t-shirt fabric and treating their exposed skin with an effective repellent, the lower forearm of a participant was divided into two portions: the distal 15 cm was treated with 20% DEET (0.33 μl/cm$^2$) and the proximal 10 cm covered with untreated cotton fabric from a white t-shirt. The product was applied via a pipette, by carefully dispensing drops across the treatment area and then spreading across the skin using a gloved finger. The second arm of the participant was left untreated and uncovered, and was used as a control. The whole of the control lower arm was introduced to a Nylon cage (30 cm x 30 cm x 30 cm BugDorm) containing 30 colony-reared, non-bloodfed, 5-7-day-old female *Aedes aegypti*. The number of probing events on the proximal and distal part of the arm was counted over 75 seconds, then the arm was removed. The count included the total number of mosquitoes that attempted to probe during this period, with the potential for a single mosquito in the cage to be counted more than once. The treated lower arm was then introduced, and the same count made. Three arm-in-cage replicates were performed. Mosquitoes were replaced and the test repeated if the number probing in the control arm was less than ten.

Protective efficacy (*PE*) was determined as a proportion of the number of mosquito probings on the treated arm (*T*) in relation to the number of probings on the control (untreated) arm (*C*):

$$PE = (C - T)/C$$

### Testing of repellents added to clothing in fabric softeners

Fabric softeners containing insect repellents are commercially available and are marketed as products that can be added to a laundry cycle to provide bite protection when the treated clothing is worn. We tested three such products containing the repellent IR3535 (MosquitoNo) or citronella oil (NoMo, Si Repel) in arm-in-cage tests as described above. Cotton fabric from

white t-shirts was washed using unscented laundry detergent in a standard 40°C wash cycle and the product added as fabric softener in the rinse cycle. The fabric was allowed to air dry before being used. Five arm-in-cage replicates were performed immediately after the fabric was dry.

## Selection of active compounds for spray treatments

An alternative to adding repellent to clothing during the rinse cycle is to apply it after washing, when the clothing is dry. This is expected to increase the efficacy of a repellent, as the active compound will not be lost during rinsing. We selected seven natural and synthetic compounds for spray treatment of clothing: 2-undecanone, DEET, deltaundecalactone (dUDL), geraniol, IR3535, methyl anthranilate (MA), and p-menthane-3,8-diol (PMD). To 20 cm x 20 cm sections of cotton fabric from a white t-shirt, 1 ml of repellent mixed with 9 ml of ethanol was added. The 10 ml solution was applied to the fabric using a small spray bottle. The active ingredients were applied at three concentrations (0.1%, 1% and 10%), and control sleeves were prepared with ethanol. The fabric was allowed to dry before being used in arm-in-cage tests as described above. In these tests, the sleeve covered the whole of the lower arm, with no skin left exposed (Fig 1A). Therefore, all probes were recorded on the fabric. Four replicates were performed for each active ingredient and concentration.

## Longevity of protection

To test the longevity of protection provided by dUDL, the active ingredient was solubilised in ethanol at four different concentrations (2%, 4%, 8% and 15%) and applied to a 25 x 25 cm (625 cm$^2$) cotton fabric sleeve. The sleeve was allowed to dry, then was used to cover the whole of the lower arm. Arm-in-cage tests were conducted as before, with an uncovered arm used as a control. The tests were performed at 0 h, 24 h, 120 h and 168 h after the treatment had dried, to represent use up to one week after application.

## Protection of covered and uncovered skin

To test the ability of treated clothes to provide protection to uncovered skin, the lower arm was conceptually divided into two. The distal part of the lower arm was left uncovered and the

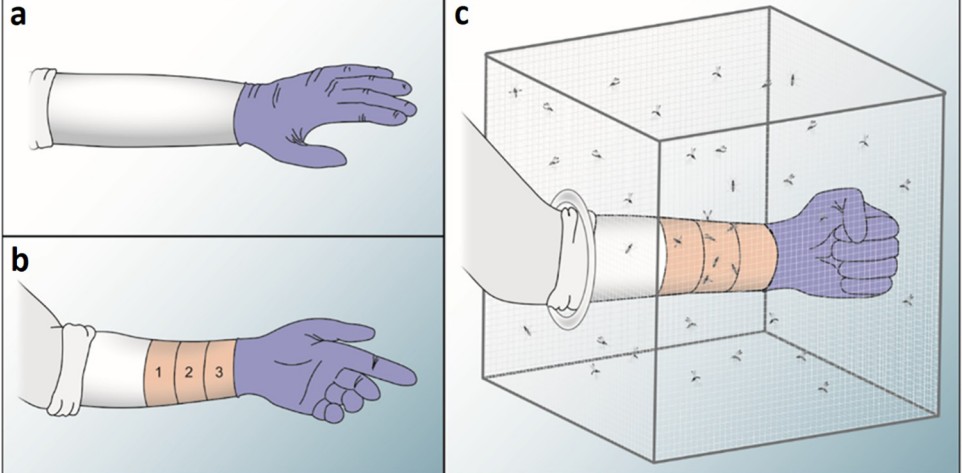

**Fig 1. Arm-in-cage tests.** a) Whole of lower arm covered with cotton fabric. b) Proximal part of lower arm covered, and distal part marked with three 5-cm divisions (0–5 cm, 5–10 cm, and 10–15 cm distal to the sleeve). c) Lower arm inside cage of mosquitoes. The fabric was secured to fit close to the skin.

proximal part covered with dUDL-treated sleeve at a concentration of 0.48 μl/cm$^2$. Protective efficacy was determined compared to a control, which used an untreated sleeve on the proximal part of the lower arm.

In addition, dUDL was applied as a topical treatment, and was used with an untreated cotton sleeve. The topical treatment was applied at a rate of 0.12 μl/cm$^2$. Finally, the treated sleeve and treated skin were tested in combination, to assess the value of a single formulation that could be applied to both the skin and clothing.

In each of these tests, the arm was marked with non-scented marker at 5 cm intervals, so that mosquitoes probing in each portion of the arm could be recorded (Fig 1B and 1C). As above, these tests were conducted over 75 seconds, and were performed with five replicates. At each hourly time point, a control test was performed followed by a treatment test, and tests continued over a period of up to 8 hours post-treatment. Tests were stopped if the PE of a test cage dropped below 50%.

## Comparison of dUDL with PMD

The set of three tests described above with treated sleeve, treated skin, and combination of treated sleeve and skin were repeated with a second repellent, PMD, to investigate whether the more novel active ingredient, dUDL, provided greater levels of protection than the more established repellent. PMD was solubilised in ethanol and applied at the same rate as that used for dUDL. Five replicates were performed for up to 8 hours post-treatment.

## Statistical analysis

A Kaplan-Meier survival analysis was conducted in SPSS (version 25, IBM Corp. 2017) to determine the mean time of complete protection for sleeve, skin, and sleeve and skin combination treatments, with both dUDL and PMD.

## Mathematical modelling

Mathematical modelling was employed to investigate the potential value of a repellent product that provided the levels of protection reported from our arm-in-cage tests. We used an existing model of arbovirus transmission dynamics for which published parameter estimates exist from different contexts for ZKV, DNV, and CHKV to simulate the number of human infections with and without a mosquito repellent [16–18]. The model represents a closed population without in- or out-migration. This means conclusions drawn from the modelling analysis apply to the population in endemic regions, not to travellers. The model definition and parameter values used for each arbovirus are given below (Tables 1 and 2).

We simulated outbreaks with two different $R_0$ values for each arbovirus resulting in outbreaks of different magnitudes and dynamics (S1 Fig). The $R_0$ values were chosen to represent the upper and lower ranges of the distributions of the estimated values for $\beta_H$ and $\beta_v$ estimated in the papers referenced in Table 2 (see S1 Table for equation of $R_0$). We assumed a mosquito-to-human ratio of 1.0 and only varied the transmission rate. Changing the mosquito-to-human ratio would be equivalent to changing the value of $R_0$. We assumed that symptomatic and asymptomatic infected humans were equally infectious to mosquitoes. We also looked at the difference in outcomes if asymptomatic infected humans were only half as infectious as symptomatic infected humans. We simulated repellent use with a range of possible hypothetical parameter values and recorded the proportion of infections averted for each repellent use scenario relative to the baseline scenario without repellent use. We did not consider any other interventions (such as vaccines or bed nets) in the simulation. The simulations were run in R (version 4.3.1, R Core Team 2023).

**Table 1. Equations of a generic arbovirus transmission dynamic model including the effect of a mosquito repellent.**

| | |
|---|---|
| Susceptible humans | $\frac{dS_H}{dt} = -\beta_H(1 - c_D D)S_H I_v$ |
| Exposed humans | $\frac{dE_H}{dt} = \beta_H(1 - c_D D)S_H I_v - \alpha_H E_H$ |
| Infected humans (symptomatic) | $\frac{dI_{HS}}{dt} = \theta\alpha_H E_H - \gamma I_{HS}$ |
| Infected humans (asymptomatic) | $\frac{dI_{HA}}{dt} = (1 - \theta)\alpha_H E_H - \gamma I_{HA}$ |
| Infected humans (cumulative) | $\frac{dC}{dt} = \alpha_H E_H$ |
| Recovered humans | $\frac{dR}{dt} = \gamma(I_{HS} + I_{HS})$ |
| Total humans | $H = S_H + E_H + I_{HS} + I_{HA} + R$ |
| Susceptible mosquitoes | $\frac{dS_v}{dt} = \delta V - \beta_v(1 - c_D D)S_v \frac{(I_{HS} + \sigma I_{HA})}{H} - \delta S_v$ |
| Exposed mosquitoes | $\frac{dE_v}{dt} = \beta_v(1 - c_D D)S_v \frac{(I_{HS} + \sigma I_{HA})}{H} - (\delta + \alpha_v)E_v$ |
| Infected and infectious mosquitoes | $\frac{dI_v}{dt} = \alpha_v E_v - \delta I_v$ |
| Total mosquitoes | $V = S_v + E_v + I_v$ |
| Repellent efficacy (decay process) | $\frac{dD}{dt} = -\frac{\log(2)}{h_D}D$ |

We further calculated the direct healthcare costs and the costs of lost labour (analysis from a societal perspective) that could be averted with a mosquito repellent with a given efficacy and half-life if a certain percentage of the population correctly applied it with a set average time between reapplications. We compared the averted costs to the total cost for mosquito repellent

**Table 2. Parameter values for arbovirus transmission model.** Values are maximum likelihood estimates from cited sources. For parameters determining the transmission rate, we selected values from the estimated range or 95% credible interval to simulate outbreaks with different values of the basic reproduction number ($R_0$).

| Parameter symbol | Parameter explanation | CHKV values | DNV values | ZKV values |
|---|---|---|---|---|
| **Human parameters** | | | | |
| $\beta_H = p_{H}\eta$ | Transmission rate from mosquitoes to humans per day (tested values) | 0.14, 0.21 [16] | 0.165, 0.3 [17] | 0.27, 0.37 [18] |
| $p_H$ | Transmission probability from mosquitoes to humans | – | 0.5 [17] | – |
| $\eta$ | Average mosquito biting rate per day | – | 0.33, 0.6 [17] | – |
| $\frac{1}{\alpha_H}$ | Incubation period in infected humans in days | 2 [16] | 2 [17] | 6.6 [18] |
| $\frac{1}{\gamma}$ | Infectious period in humans in days | 4 [16] | 4 17 | 5.6 [18] |
| $\theta$ | Proportion symptomatic in humans | 0.83 [16] | 0.45 (for primary infections) [19] | 0.5 [20] |
| $\sigma$ | Infectiousness of asymptomatic infections relative to symptomatic (assumption) | 1.0 | 1.0 | 1.0 |
| **Mosquito parameters** | | | | |
| $\beta_v = p_v\eta$ | Transmission rate from humans to mosquitoes per day | 0.2, 0.4 [16] | 0.165, 0.3 [17] | 0.27, 0.37 [18] |
| $p_v$ | Transmission probability from humans to mosquitoes | – | 0.5 17 | – |
| $\frac{1}{\alpha_v}$ | Incubation period in infected mosquitoes in days | 2 [16] | 6 [17] | 6.3 [18] |
| $\delta$ | Mosquito death/birth rate per day | 1/20 [16] | 1/23 [17] | 1/12 [18] |
| **Repellent parameters (tested values same for all arboviruses, assumptions)** | | | | |
| $D_0$ | Initial efficacy (directly after correct application) | 0.25, 0.5, 0.75, 1.0 | | |
| $h_D$ | Half-life | 6 hours, 12 hours, 1 day, 2 days | | |
| $c_D$ | Coverage (proportion of human population applying repellent) | 0.1, 0.2, 0.3, 0.4, 0.5, 0.6, 0.7, 0.8, 0.9, 1.0 | | |
| $t_D$ | Time between reapplications of repellent | 4 hours, 6 hours, 8 hours, 12 hours, 1 day, 2 days, 3 days, 4, days, 5, days, 6, days, 7 days | | |

that was required to avert these costs. The cost for the repellent was calculated as follows:

$$cost_{repellent\ total} = \frac{T}{t_D} c_D S_{H,0} cost_{repellent\ application}$$

Where $T$ is the duration of the outbreak, $t_D$ the time between repellent applications, $c_D$ is the proportion of the human population applying the repellent (coverage) and $S_{H,0}$ is the total human population.

The costs that can be averted by a given repellent regimen are disease-specific. For ZKV, we considered outpatient medical costs for mild symptomatic disease, and hospitalisation costs for Guillain-Barré Syndrome (GBS) and for microcephaly, as well as lost labour due to mild symptomatic disease and GBS. For CHKV we considered outpatient costs for acute-phase mild symptomatic disease, hospitalisation costs for acute-phase severe disease, medical costs for chronic disease and lost labour due to symptomatic disease during the acute and chronic phase up to twelve months following infection. For DNV we considered the medical costs for outpatients (mild disease) and hospitalised patients (severe disease) and lost labour due to either mild or severe disease. Assumptions and parameter values used in the cost calculations are detailed in S1 Table. Cost values are for the Latin American and Caribbean context. Equations for each cost calculation are presented in S1 Table. All costs extracted from original sources have been converted to 2023 US dollars using World Bank data on inflation and GDP per capita (August 2023) [21]. Discounting had been applied to costs in the source publications. No potential harms from repellent use were considered.

From the cost averted, we can calculate the maximum cost per repellent application for which a repellent can be considered cost-effective, meaning the average repellent cost per individual is less than or equal to the value of the benefits from repellent use (cost neutral at the population level). We calculate this cost assuming a coverage of 100% (ideal scenario) so that the total costs and benefits of the intervention are evenly spread across the population:

$$cost_{repellent\ application\ max} = \frac{t_D}{Tc_D} \left( cost_{HC,averted} + cost_{lost\ labour,averted} \right)$$

Where $cost_{HC,averted}$ is the healthcare cost averted by a given repellent regimen and $cost_{lost\ labour,averted}$ is the averted cost of lost labour caused by disease. In addition, we calculate the cost for 80% coverage (high-coverage scenario), 40% coverage (medium-coverage scenario) and 10% coverage (low-coverage scenario). If coverage is less than 100%, costs for repellent protection are not evenly spread across the population.

## Results

### Demonstration of probing on covered skin

Arm-in-cage tests with a participant arm partially covered with cotton fabric from a white t-shirt and partially treated with DEET found that the DEET fully prevented mosquito probings where it had been topically applied, but did not provide protection to the untreated, fabric-covered skin (S2 Fig). This initial study demonstrated the need to provide protection from mosquitoes that extends beyond the application of a highly effective repellent applied to exposed skin.

### Application of repellents to clothing in fabric softeners

Three commercially-available fabric softeners containing insect repellents failed to provide a high degree of protection from mosquitoes. The highest mean PE determined from five

replicates was 11.2% compared to non-treated fabric controls (S3 Fig). No further investigations were made with wash-in repellents.

## Selection of active compounds for spray treatments

Seven active ingredients were investigated for their ability to provide protection to clothing when sprayed directly onto cotton fabric as a post-wash treatment. Arm-in-cage tests with the cotton fabric used as sleeves to cover the whole of the lower arm revealed that 100% protection could be achieved once the fabric had dried; no mosquitoes were recorded probing on the sleeves treated with 2-undecanone, DEET, dUDL, geraniol, IR3535, MA or PMD at the highest tested concentration (10%). Protection provided by PMD and 2-undecanone dropped to less than 80% when applied at 1%, while DEET, geraniol, IR3535 and MA retained complete protection at 1%. No compounds provided complete protection at 0.1%, although DEET and geraniol were able to provide an average of greater than 75% protection (Fig 2). However, it was noted during these tests that several of the active ingredients, particularly when used at the higher concentrations, had unpleasant or unnatural smells, and would be unlikely to be acceptable to users according to the preferences stated in focus group studies. As a result of the demonstrated efficacy and pleasant fruity smell, dUDL was selected for further investigation.

## Longevity of protection

Up to 96% protection after 168 hours was provided by the highest concentration of dUDL in ethanol applied to clothing that covered the whole of the lower arm (Fig 3).

## Protection of covered and uncovered skin

The ability of dUDL-treated clothing to provide protection to uncovered skin would rely on a spatial effect of the active ingredient. Arm-in-cage tests with lower arms only partially covered with treated cotton fabric (and the rest of the lower arm uncovered) revealed that treatment of the sleeve alone provided very little bite protection: at the time of application, 46%

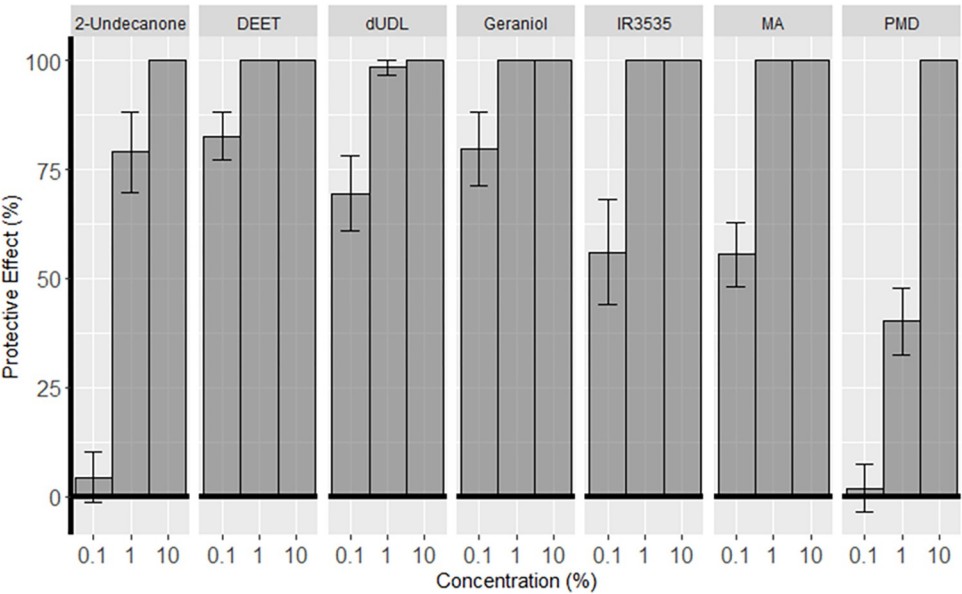

**Fig 2. Mean protection (±S.E. bars) provided by seven active ingredients tested at three concentrations in arm-in-cage tests at t = 0.** n = 4 replicates.

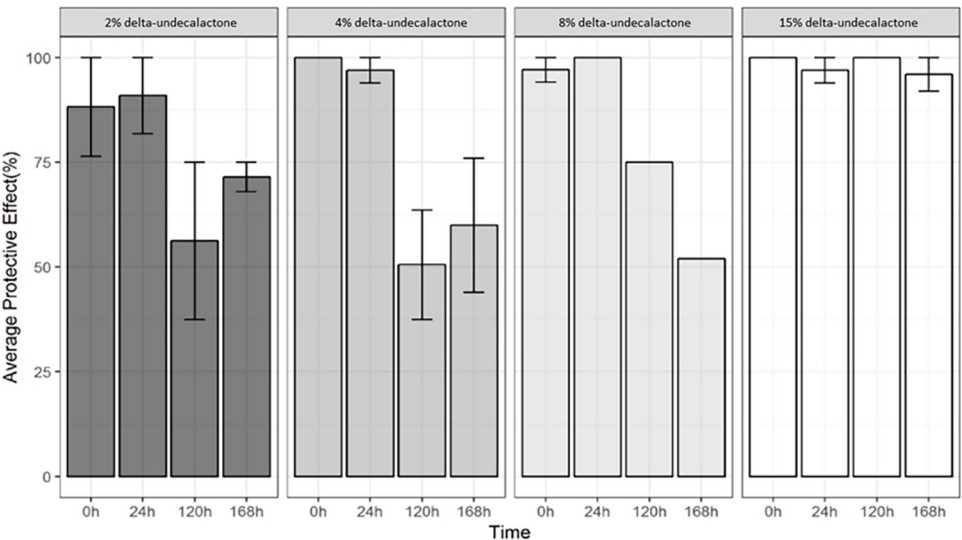

**Fig 3. Mean protection (±S.E. bars) provided by dUDL at times points up to 168 hours.** Four concentrations were tested in arm-in-cage tests. n = 2 replicates.

(S.E. ±7.3%) protection was provided. When the treatment was instead applied to the skin and not the clothing, the protection provided was 93% (S.E. ±3.1%). In a third set of tests, the treated cotton sleeve was coupled with topical treatment. Here, the protection was found to be 95% (S.E. ±3.4%) at the first time point, and remained above 50% for 7 hours (Fig 4).

In the tests without a topical treatment, where only the cotton sleeve was treated, counts of mosquitoes probing in each portion of the arm revealed a high level of protection to the covered part of the arm; no mosquitoes probed the cotton sleeve (Fig 5).

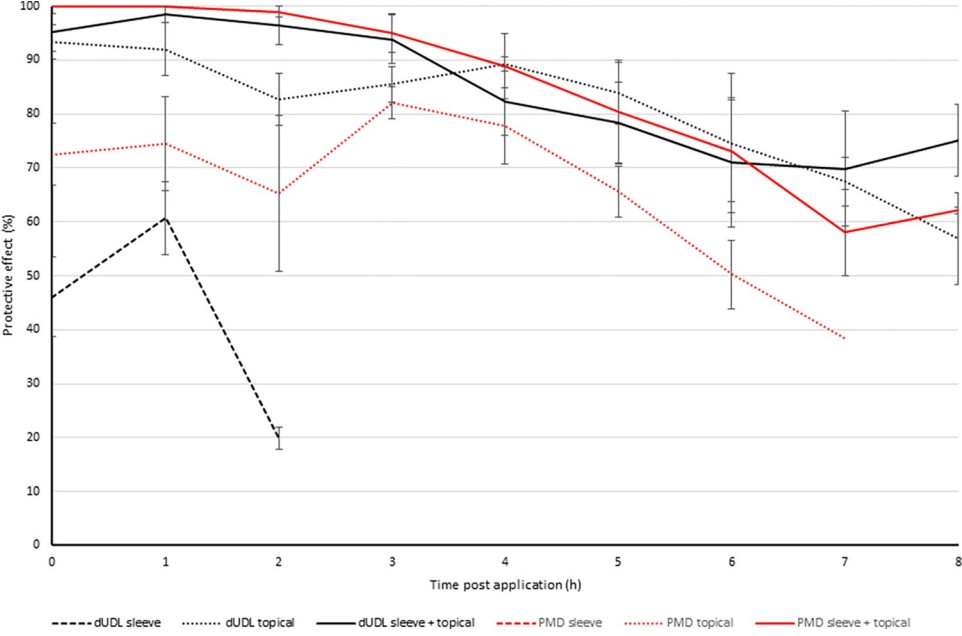

**Fig 4. Mean protective effect (± S.E. bars) provided by dUDL and PMD treatments over time.** Protective efficacy calculated against an untreated, uncovered control arm. n = 5 replicates. Note that if P.E. in any given cage dropped below 50%, testing with that cage was not continued, so not all data points represent the mean of 5 replicates.

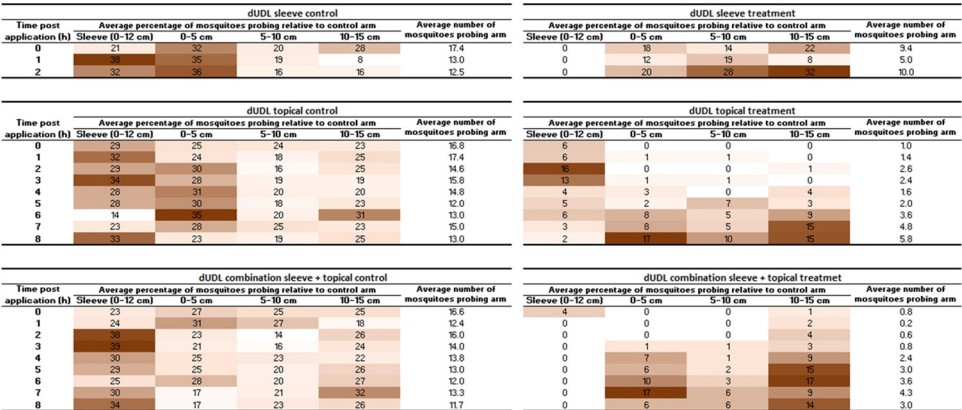

**Fig 5. Distribution of probing mosquitoes on sections of arm.** Left panel shows distribution on the control arm, calculated as the average number of mosquitoes probing each section relative to the average number that probed across the whole of the lower arm in control tests, presented as a percentage. Right panel shows the same for the dUDL treated arm, also relative to the control tests. For example, in the control tests conducted for the sleeve treatment, an average of 17.4 mosquitoes probed across the lower arm at 0 hours post application. Of these, an average of 5.6 (32%) probed in the 0–5 cm distal portion of the arm. In the tests with the dUDL sleeve treatment, an average of 3.2 (18%) probed the same portion. n = 5 replicates. Darker colours indicate a larger proportion of mosquitoes probing in that portion of the arm.

Counts of mosquito probing in the tests with an untreated sleeve and dUDL applied to the skin showed the opposite distribution of mosquitoes (Fig 5). The distal part of the arm initially received no probings, and the mosquitoes recruited to the sleeve. This observation is consistent with our findings with DEET, described above, which demonstrated the need to protect skin that is covered only by untreated clothing. Over time, as the repellent lost efficacy, more mosquitoes were attracted to the uncovered part of the arm and eventually very few probed the sleeve.

Finally, when the dUDL-treated sleeve was paired with treatment of the skin the highest level of protection was found, particularly for the first three hours. The few mosquitoes in the test cage that probed were recorded closest to the wrist and furthest from the treatment on the sleeve. Over time, more mosquitoes probed the distal part of the lower arm, until product failure when PE was <50%.

## Comparison of dUDL with PMD

Tests of protection provided to covered and uncovered skin were repeated using PMD. Arm-in-cage tests with lower arms only partially covered with treated cotton fabric revealed that treatment of the sleeve alone provided very little bite protection: at the time of application, 27% (S.E. ±8.4%) protection was provided. When the treatment was instead applied to the skin and not the clothing, the protection provided was 72% (S.E. ±5.7%). In a third set of tests, the treated cotton sleeve was coupled with topical treatment. Here, the protection was found to be 100% (S.E. ±0%) at the first time point, and remained above 50% for 8 hours (Fig 6).

The position data provided similar results to those seen with dUDL. When the treated sleeve was tested alone, without a topical treatment, it did not receive any probing mosquitoes. Interestingly, the mosquitoes probed closer to the wrist and away from the PMD-treated sleeve. When PMD was used as a topical treatment alone, no mosquitoes initially landed on the treated, uncovered, skin. However, as with dUDL topical treatment, more mosquitoes began to probe the uncovered arm over time, and the product failed after 7 hours. In the combination sleeve and topical treatment, no mosquitoes were observed probing until the 2 hour time-point, and none probed on the treated sleeve over the full 8 hours of testing.

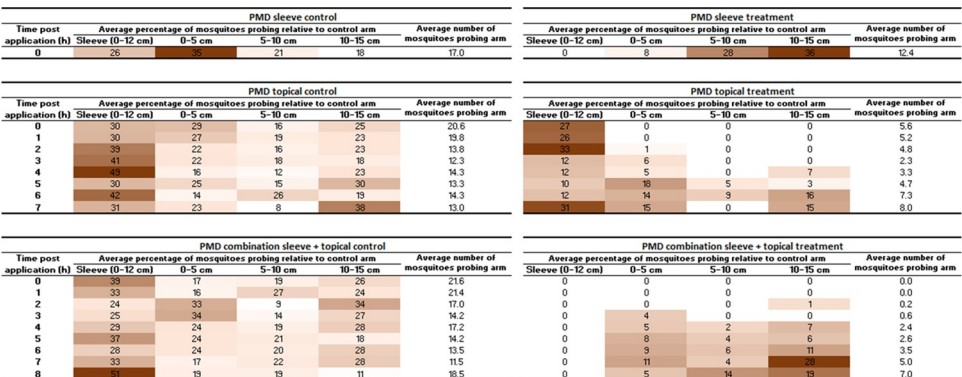

**Fig 6. Distribution of probing mosquitoes on sections of arm.** Left panel shows distribution on the control arm, calculated as the average number of mosquitoes probing each section relative to the average number that probed across the whole of the lower arm in control tests, presented as a percentage. Right panel shows the same for the PMD treated arm, also relative to the control tests. n = 5 replicates. Darker colours indicate a larger proportion of mosquitoes probing in that section.

Kaplan-Meier survival analysis was used to determine the mean and median protection time provided for each product, representing the time point at which 50% of mosquitoes in the cage were probing. For dUDL, when applied only to the sleeve, mean protection time was determined to be 52 minutes (Table 3). This compared with just 6 minutes for PMD applied to the sleeve. When the treatments were applied directly to the skin, the mean protection time increased to 7 hours 37 minutes and 5 hours 12 minutes, respectively. The greatest protection was provided by the dUDL applied as sleeve and topical treatments, averaging 7 hours 33 minutes. In the same test format, PMD provided complete protection for 7 hours 12 minutes.

## Modelling repellent-based prevention

Modelling suggests that a substantial proportion of arbovirus infections and symptomatic cases can be prevented by a durable and highly effective repellent, provided that a sufficient proportion of the population apply it at a high enough frequency (Figs 7 and S4). In our simulation analysis, a repellent with 100% initial efficacy after application and a half-life of 7 hours (as found for dUDL with our survival analysis) can avert more than 30% of ZKV infections (outbreak $R_0 = 2.2$) if it is applied twice per day by at least 80% of the population (Table 4). In a less severe outbreak ($R_0 = 1.6$) the same repellent regimen can avert 96% of infections. In other simulated scenarios, the results were similar for the same repellent regimen: for DNV outbreaks with $R_0 = 2.6$, 26.6% infections would be averted, and for DNV outbreaks with $R_0 = 1.4$, 98% of infections would be averted. For CHKV, outbreaks with $R_0 = 2.2$, 31.6% of infections would be averted, and for CHKV outbreaks with $R_0 = 1.4$, 99% of infections would be averted.

**Table 3. Kaplan-Meier analysis to determine mean protection time of dUDL and PMD in three treatment formats.** *Not known.

| Active ingredient | Treatment | Mean time at which PE was <50% (hh:mm, 95% CI) | Median time at which PE was <50% (hh:mm, 95% CI) |
|---|---|---|---|
| dUDL | Topical | 07:37 (06:54–08:19) | >08:00 |
| | Sleeve | 00:52 (00:00–01:51) | 00:07 (00:02–00:11) |
| | Combination sleeve + topical | 07:33 (06:48–08:17) | >08:00 |
| PMD | Topical | 05:12 (03:25–06:58) | 05:58 (01:44–10:11) |
| | Sleeve | 00:06 (00:05–00:07) | 00:06 (NK*) |
| | Combination sleeve + topical | 07:12 (06:12–08:12) | >08:00 |

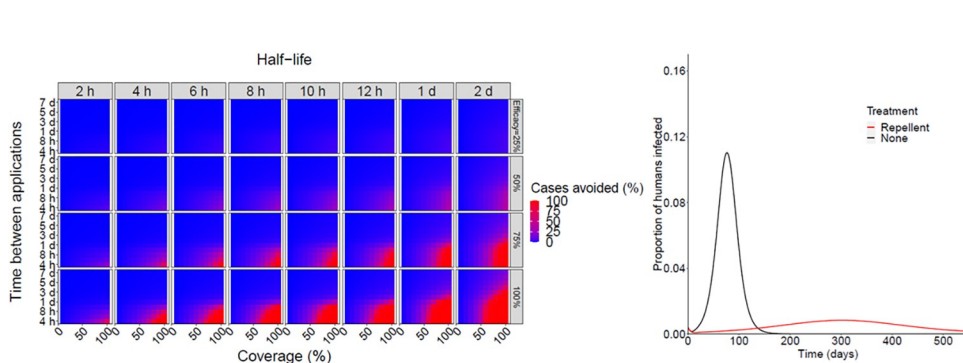

**Fig 7. Impact of repellent on ZKV infections during an outbreak.** a) Heatmap showing proportion of infections and symptomatic cases avoided over 18 months depending on repellent coverage (x-axis), time between applications (y-axis), repellent efficacy (secondary y-axis) and half-life (secondary x-axis). The more red colouring, the more cases can be prevented by a repellent with the given characteristics. b) Example of epidemic curves with and without repellent. Repellent coverage 80%, application frequency: 1x every 12 hours, efficacy 100%, half-life 7 hours.

High population-level benefit can only be achieved for high coverage of repellent use in the population. If a much lower proportion of the population adopted the same intervention, the percentage of infections averted would be considerably lower. If 40% of people used the repellent, 4.4% of ZKV (outbreak $R_0$ = 2.2), 3.3% of DNV (outbreak $R_0$ = 2.6) and 4.6% of CHKV (outbreak $R_0$ = 2.2) infections would be averted (Table 4).

If asymptomatic infected individuals are only half as infectious as symptomatic infected individuals, the same repellent regimen can avert relatively more infections in our simulations: 50% of ZKV infections in an outbreak with $R_0$ = 2.2, 98% of ZKV infections in an outbreak with $R_0$ = 1.6, 48% of DNV infections in an outbreak with $R_0$ = 2.6, 98% of DNV infections in an outbreak with $R_0$ = 1.4, 51% of CHKV infections in an outbreak with $R_0$ = 2.2, and 99% of CHKV infections in an outbreak with $R_0$ = 1.4 (see S4 and S5 Figs).

In our CEA simulations, a repellent with 100% initial efficacy and twice daily application by 80% of the population was estimated to avert total healthcare costs per person (entire susceptible population) of $369 for CHKV, $26.07 for DNV, $14.40 for ZKV. The costs per person due to lost labour averted by this regimen are $70.30 for CHKV, $28.93 for DNV, $5.22 for ZKV. For a city with about 2 million inhabitants like Manaus in Brazil, the total costs for healthcare and lost labour averted would be $879.6 million during a CHKV outbreak, $110 million during a DNV outbreak, $38.5 million during a ZKV outbreak. The higher cost averted for CHKV compared to DNV and ZKV derives from our consideration of costs for chronic CHKV disease. The costs averted for ZKV disease account for life-time healthcare costs for microcephaly but not for costs of lost labour for carers of microcephalic children because of lack of data.

**Table 4. Percentage of cases avoided by different levels of intervention coverage in severe outbreak scenarios (ZKV outbreak $R_0$ = 2.2, DNV outbreak $R_0$ = 2.6, CHKV outbreak $R_0$ = 2.2).** Assumes repellent has 100% initial efficacy after application and a half-life of 7 hours, with two applications per day, and same infectiousness in symptomatic and asymptomatic infected individuals.

| Arbovirus | Population coverage | | | |
|---|---|---|---|---|
| | Low (10%) | Medium (40%) | High (80%) | Ideal (100%) |
| Dengue | 0.4 | 3.3 | 26.6 | 88.3 |
| Chikungunya | 0.6 | 4.6 | 31.6 | 90.8 |
| Zika | 0.5 | 4.4 | 32.1 | 91.8 |

**Table 5. Maximum cost per application (in US dollars) at which repellent can be considered cost-effective by different levels of intervention coverage in severe outbreak scenarios (ZKV outbreak R0 = 2.2, DNV outbreak R0 = 2.6, CHKV outbreak R0 = 2.2).** Assumes repellent has 100% initial efficacy after application and a half-life of 7 hours, with two applications per day.

| Arbovirus | Population coverage | | | |
|---|---|---|---|---|
| | Low (10%) | Medium (40%) | High (80%) | Ideal (100%) |
| Dengue | 0.0066 | 0.015 | 0.063 | 0.17 |
| Chikungunya | 0.073 | 0.15 | 0.50 | 1.15 |
| Zika | 0.0029 | 0.0062 | 0.022 | 0.051 |

The maximum cost per application at which a repellent can be considered cost-effective, assuming a half-life of seven hours an application frequency of two times per day, is given in Table 5. The higher effective cost per application for CHKV is driven by chronic CHKV disease. The maximum allowable cost per application is up to 25 times smaller for low coverage (10%) than for 100% coverage, reflecting the low population-level benefits achieved by low repellent coverage.

## Discussion

*Aedes*-borne diseases continue to pose a threat to populations in those climates that support the mosquito lifecycle. Whilst several interventions have been developed and applied for the control of *Aedes* [22], additional tools are needed, particularly for people outside their home and away from the protection offered by window and door screens. Insect repellents applied to the skin are the most familiar option for personal use, but are limited by variable user compliance, cost and how long they last. Studies of both short-term users, such as those taking vacations in endemic areas, and long-term or routine users in such areas indicate that usage rates can be low [23,24]. For example, in Cambodia, a 20-month cluster-randomized trial on the effectiveness of topical repellents in controlling malaria infections at community level reported daily use of repellent to be just 8%, despite widespread availability [25]. According to our simulation results, such low coverage would avert less than 1% of total infections. A trial in Laos estimated full compliance of repellent use in the range of 48–60% over an eight-month period [26]. Indeed, compliance emerges as one of the most decisive factors for the success of interventions with repellents [26]. Reasons for not using topical applications include forgetting, disliking the smell, allergies, safety fears, and costs [14,27]. Further, in warm climates, perspiration of the skin may demand that repellents are applied more frequently, and there may be reluctance of people to adapt to the routines of applying such products [25,28]. Our simulations show that with a half-life of 7 hours, twice daily application could be sufficient to prevent a substantial proportion of cases provided coverage is high enough. In this context, it is crucial to research user preferences, optimize products to ensure they meet the requirements of targeted communities, and place minimal reliance on human behaviours.

We aimed to produce wearable clothing that would provide a high degree of protection from *Aedes* bites. If we consider the most effective intervention to be that which is the most accepted and is, therefore, met with proper and consistent usage [29], it was also important that our target product could be applied to the clothing of the wearer's choice. The alternative, clothing that is pre-treated and perhaps only available in a limited range of designs, was not favoured by participants of our focus groups [14]. Ideally, clothing would be treated at the time of laundering through a fabric softener, so once applied, daily use of the repellent would not require any deviation from daily routine. However, our testing with repellent additives in fabric softener revealed the difficulties in ensuring a sufficient retention of active ingredient to provide protection when the clothing was dried and worn. Our attention turned to spray-on

application of fabrics. This approach has been tested elsewhere [30,31], and given the lack (and need) for innovation in wash-in repellent formulations, offers an appealing simple alternative to wash-in repellents.

Delta-undecalactone as a spray-on repellent that can be applied to clothing offers a new choice of PPT. Although synthetic, dUDL is found in nature and has a fragrance described as fruity. It can be used to add peach, apricot or vanilla and cream notes when used for flavouring [32], and has not been shown to induce skin sensitisation in volunteers [33]. It is available at low cost and when used at high concentrations, dUDL may provide protection after several days. However, whilst a spray-on product may prevent bites through thin clothing, we are cautious to note that treatment to clothing that does not fully cover the skin leaves the wearer vulnerable to mosquito bites on exposed areas. A theme emerging from our focus group discussions concerned a desire to wear short-legged and short-armed clothing in the day-time heat. A textile spray would only provide protection to a limited portion of the body, and full protection would require the additional application of a topical repellent.

Protection to uncovered skin could only be provided by active compounds with spatial effects. Such properties have been reported for a number of insecticides, and there is growing understanding of the non-lethal effects of such compounds on mosquitoes [34–36]. For example, DDT applied inside huts creates a spatial repellent barrier that deters a large proportion of available mosquitoes from entering, contributing to reducing human-vector contact without the toxic effect of the insecticide [37]. By contrast, permethrin is a fast-acting contact insecticide [38], meaning that when it is applied to clothing it can irritate or kill insects before they feed, but it is not expected to provide protection to uncovered skin. Similarly, the repellents DEET, picaridin and IR3535 are classified as contact repellents, and require mosquitoes to come into close proximity or direct contact with a treated surface in order to be repelled [39].

The essential oils of such plants as lemongrass and cinnamon have been shown to provide protection from *Aedes* mosquitoes through spatial effects [40]. Although the mode of action of dUDL has not been described, it has been shown to act as a spatial repellent that has an effect over a large distance [41]. Volatilization of spatial repellent compounds creates repellent vapours that come into contact with host-seeking mosquitoes and elicit an aversive behaviour or deleterious physiological response [42]. The present study has shown that the low concentration of dUDL applied to clothing has a limited spatial effect. Our tests with treatment on the sleeve suggested a possible spatial effect of PMD (Fig 6), but because the protective efficacy was less than 50%, testing was completed after just one time point; we were not able to see whether this effect deteriorated over time, indicated by progressively more probing closer to the sleeve.

In order to provide a high level of bite protection to uncovered skin, our results suggest that either a higher concentration of dUDL would be needed, perhaps exceeding that tolerated by consumers with regards to the odour the clothing would produce, or that the treated clothing should be paired with a topical treatment. We recognise that a combination product does not overcome the issues of low compliance that limits the utility of topical repellents. The introduction of such a product may require some education component to instil a recommendation that community members treat their clothing to prevent bites to their covered skin, and apply the product to their skin to ensure the highest levels of protection. Communication and behaviour change strategies would be required to achieve the highest levels of coverage and the highest proportion of cases averted that our models describe.

Modelling offers the opportunity to explore the potential effects of interventions that have not yet been developed but which might represent target product profiles, or to investigate what might be achieved if known products were to be more widely adopted due to communication campaigns. We have produced a flexible model that allows for different protective efficacies, half-lives, time between applications and population coverage to be independently

altered and the impacts on cases of arboviral infections to be determined in outbreaks of different severity. We show that it is not necessary to have 100% population usage to have community-wide benefits, but also, at the opposite end of the spectrum, if usage is very low then the number of cases averted remains small. Evidence from the literature suggests that achieving high coverage of repellent usage is difficult in practice, although high coverage has been reported and confirmed in intervention trials [25,43]. The protective effect of repellent at the population level increases non-linearly with coverage, in a similar way to herd immunity from vaccination. However, whereas vaccination-induced herd immunity can be long-lasting, repellent needs to be reapplied frequently over an extended period of time to maintain its protective effect. On the other hand, repellent can protect against multiple arboviral diseases and can, therefore, be a valuable complement to other anti-arboviral interventions. Efficacy of a repellent needs to be high (75% or higher) to realistically achieve a noticeable impact on the number of infected cases. Longer half-life of a repellent or higher reapplication frequency among users can make up for lower population coverage. We have shown that dUDL is able to achieve this high level of efficacy, at least initially, and other more established products such as DEET-based repellents are expected to meet the target product profile for a PPT that would effectively reduce the burden of infection during an outbreak if population coverage was sufficiently high. However, we recognise that ultimately the dUDL-based tools we have investigated would require users to apply the product to their clothing, and there are practical limitations related to this. We also note that our study has used laboratory-reared mosquitoes, which may respond differently the test substances to those encountered in the wild.

With regard to product costs, the typical price per bottle of icaridin-based mosquito repellent in Brazil is quoted as $4.19–10.74 in 2019 ($5.25–12.51 in 2023 US$) [44]. Assuming one bottle lasts for 50 applications, the cost per application is just $0.105–0.250. Future dUDL or other clothing- and skin-based repellents could be priced at this or a higher bracket and still remain a cost-effective intervention to prevent ill health from arboviral diseases. Note that our analysis only considered the population endemic regions. The threshold for cost-effectiveness in travellers may be different. In our analysis, repellent is most likely to be cost-effective for CHKV infection, in terms of healthcare cost and lost labour averted. This is mainly driven by the cost of long-term sequelae following CHKV infection. If costs for long-term sequelae of DNV and ZKV infections were better documented, the cost-effectiveness of repellent for these diseases would be likely to increase further [45]. In our calculations we assumed that all individuals in the population were equally likely to be bitten by mosquitoes. In reality, some demographic groups may be more exposed to mosquito bites than others, for example, lower-income groups which may at the same time have a lower willingness-to-pay for repellents or other protective measures [44,46]. Subsidising mosquito repellents for these groups may be considered to improve coverage and application frequency.

## Supporting information

**S1 Fig. Epidemic curves for arbovirus outbreaks with different R0 values and no interventions (no repellent use).** Parameters as in Table 2. Final proportion of humans infected at end of outbreak: ZKV, R0 = 2.2–98.7%; ZKV, R0 = 1.6–92.6%; DNV, R0 = 2.6–99.1%; DNV, R0 = 1.4–76.6%, CHKV, R0 = 2.2–99%, CHKV, R0 = 1.4–78.2%.
(DOCX)

**S2 Fig. Mean protection (±S.E. bars) provided by 20% DEET topically applied to the distal part of the lower arm.** The proximal part of the lower arm was covered with untreated cotton fabric. n = 3 replicates. Protective efficacy was determined as a proportion of the number of mosquito probings on the treated arm in relation to the number of probings on the control

arm.
(DOCX)

**S3 Fig. Mean protection (±S.E. bars) provided by three commercially-available laundry additive repellents.** Efficacy was determined by arm-in-cage tests using treated fabric that covered the whole of the lower arm. MosquitNo contained the repellent IR3535, and NoMo and Si Repel contained citronella oil. n = 5 replicates.
(DOCX)

**S4 Fig. Heatmap for all arbovirus outbreak scenarios.** No difference in infectiousness between symptomatic and asymptomatic individuals.
(DOCX)

**S5 Fig. Heatmap for all arbovirus outbreak scenarios.** Assuming asymptomatic individuals are half as infectious as symptomatic individuals.
(DOCX)

**S1 Table. Assumptions used in calculations of healthcare costs and cost of lost labour avoided by repellent use in 2023 US dollars.** *Based on 5.52 million pregnancies per year in Brazil (population 214.3 million). Equations for cost calculations. Equations were customised depending on available data. Where possible cost per case (mild, severe, etc.) was used which already considers average duration of disease.
(DOCX)

## Acknowledgments

Fig 1 was produced by SciArtWork.

## Author Contributions

**Conceptualization:** Robert T. Jones, Thomas H. Ant, James G. Logan.

**Data curation:** Scott J. Tytheridge, Elizabeth A. Pretorius.

**Formal analysis:** Robert T. Jones, Carolin Vegvari, Hannah R. Meredith, Thomas H. Ant.

**Funding acquisition:** James G. Logan.

**Investigation:** Robert T. Jones, Scott J. Tytheridge, Carolin Vegvari, Hannah R. Meredith, Elizabeth A. Pretorius, Thomas H. Ant.

**Methodology:** Robert T. Jones, Carolin Vegvari, Hannah R. Meredith, Thomas H. Ant, James G. Logan.

**Project administration:** Scott J. Tytheridge, Elizabeth A. Pretorius.

**Resources:** Scott J. Tytheridge, Elizabeth A. Pretorius.

**Supervision:** Robert T. Jones, James G. Logan.

**Validation:** Thomas H. Ant.

**Visualization:** Robert T. Jones, Scott J. Tytheridge, Carolin Vegvari.

**Writing – original draft:** Robert T. Jones.

**Writing – review & editing:** Robert T. Jones, Scott J. Tytheridge, Carolin Vegvari, Hannah R. Meredith, Elizabeth A. Pretorius, James G. Logan.

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
