## [Decision Letter · Decision Letter 0]

19 Sep 2024

Dear Dr Jones,

Thank you very much for submitting your manuscript "The effectiveness of putative wearable repellent technologies to protect against mosquito biting and Aedes-borne diseases, and their economic impact." for consideration at PLOS Neglected Tropical Diseases. As with all papers reviewed by the journal, your manuscript was reviewed by members of the editorial board and by several independent reviewers. The reviewers appreciated the attention to an important topic. Based on the reviews, we are likely to accept this manuscript for publication, providing that you modify the manuscript according to the review recommendations. 

Sincerely,

Nigel Beebe, PhD

Section Editor

Nigel Beebe

Section Editor

Reviewer's Responses to Questions

**Key Review Criteria Required for Acceptance?**

**Methods**

-Are the objectives of the study clearly articulated with a clear testable hypothesis stated?

-Is the study design appropriate to address the stated objectives?

-Is the population clearly described and appropriate for the hypothesis being tested?

-Is the sample size sufficient to ensure adequate power to address the hypothesis being tested?

-Were correct statistical analysis used to support conclusions?

-Are there concerns about ethical or regulatory requirements being met?

Reviewer #1: The methods are clearly written and understood. 

L25: Can the authors provide further information about how probing mosquitoes were counted? Aedes aegypti are known to be skittish and probe the skin for a short amount of time, leave and the return to recommence probing. Are counts the total number of mosquitoes that land on the arm during the 75 second period with the potential for a single mosquito in the cage to be counted more than once?

L91: Can the authors provide further information on how the repellent was applied to skin. Note that the amount of product was applied at 0.33ul/cm2, however was this applied in a spray, cream or roll on application? This information is important in order for the reader to understand and draw conclusions on whether application technique compliance would be reasonably expected by the public. 

L119: Can the authors provide further information on how the cotton fabric was treated with the spray on repellent treatments. The authors state that 1ml of repellent is mixed with 9ml of ethanol. Is the entire 10ml of mixed formulation applied to the fabric? How is it applied? If not all of the 10ml of mixed product was applied, how do the authors ensure consistency of application?

L94: Can the authors please provide more information on the colony of Aedes aegypti used? How were the mosquitoes reared? Were they starved of sucrose prior to the probing tests? Are they known to carry insecticide resistance mutations? What is the colonies origin and how many generations has it been kept in colony? 

Figure 1: When the arm is covered with cotton fabric, is this fabric skin tight against the skin and secured? Please provide more information on how the arm is covered in fabric. 

L90: Can the authors provide information on the human participants? Was there only one human participant for all probing assays or are different personnel used? Was human ethics approval sought for the use of human participants in the probing assays? Please provide further information.

Reviewer #2: See below

**Results**

-Does the analysis presented match the analysis plan?

-Are the results clearly and completely presented?

-Are the figures (Tables, Images) of sufficient quality for clarity?

Reviewer #1: The analysis in this manuscript is adequate for the aims of the paper. The results are clearly presented and all relevant data is available to the reader. Request the authors consider the use of different coloured lines to clearly present results in Figure 4 (suggest black for dUDL and red for PMD for example as used in Figure 7b). Nil other comments regarding presentation of results.

Reviewer #2: See below

**Conclusions**

-Are the conclusions supported by the data presented?

-Are the limitations of analysis clearly described?

-Do the authors discuss how these data can be helpful to advance our understanding of the topic under study?

-Is public health relevance addressed?

Reviewer #1: Conclusions are well supported by the data presented however more discussion is required on the practicality and limitations of the public applying the repellents in order to achieve high protection rates. More information on the application methods in the methods section will provide information to the reader on the level of protection afforded by imperfect application. The authors should also discuss the potential limitations of the study given it was performed with colony-reared Aedes aegypti.

Reviewer #2: See below

**Editorial and Data Presentation Modifications?**

Reviewer #1: The paper is clearly written and easily understood by the reader. Nil significant editorial errors were found.

Reviewer #2: See below

**Summary and General Comments**

Reviewer #1: The manuscript "The effectiveness of putative wearable repellent technologies to protect against

mosquito biting and Aedes-borne diseases, and their economic impact" is an interesting, well written paper providing useful and interesting information on wearable repellent technologies. The paper is easily understood and the data is well presented. The authors should ensure that the methods are detailed enough so that the reader can understand exactly how repellents were applied. This allows the reader to draw conclusions on the applicability of these results in a public health setting.

Reviewer #2: Dear editor/authors. 

Thank you for the opportunity to read/review the manuscript titled: "The effectiveness of putative wearable repellent technologies to protect against mosquito biting and Aedes-borne diseases, and their economic impact".

I was impressed by the quality of the thinking/analyses/writing that went into the work. 

However, I struggled with the breadth of the effort and indeed wonder whether is might be better represented if provided as two publications: one on modelling and another on repellents testing. Suggested as, at least for me, the modelling efforts lose some sharpness because of the repellents testing discussion and vice versa. To be clear, this is an observation for consideration and not a recommendation. 

General comments: 

Text could/should be added to indicate that appropriate authorizations/ethics approvals were in place to support the research (or alternatively that these were not required). This point raised as, at least in North America, research authorizations/approvals often are required to test repellents with human subjects and/or for unlicensed/unregistered compounds. 

The cost effectiveness analyses (CEA) approach and related discussions could be improved/expanded (see below comments for more detail). For example, was discounting applied, were potential repellent harms considered (time to apply, health effects. 

Specific comments (by line number):

Lines 20-44 (Abstract/Author Summary). 

- "...protected form infectious bites when outdoors". Such protection might also be warranted indoors esp. for important Aedes vectors such as Ae. aegypti. 

- I read the abstract before the rest of the text and wondered (aloud) about the range of values modelled for repellent performance, e.g., the values cited in the abstract are on the "best case" side of things with respect to user uptake. Of course, further reading highlighted that the authors generated estimates with a comprehensive range of values. The abstract would (IMO) be improved by including a better representation of the entirety of the range of model inputs considered and the generalities that emerge from this, i.e. the crucial importance of user uptake. 

- "...repellents can be considered as highly cost-effective"...add appropriate caveats/context. For example, something like "...our modelling work provides support to the concept that effective repellents, i.e. that are used regularly and appropriately, could be cost-effective interventions...". 

Lines 59-65: Discussion of vaccines should be tidied up. For example:

- YFV is a LAV and is reactogenic with the potential for (rarely) severe/life-threatening AE. Suggest avoiding the trigger words such as “safe” and rather indicate it is an effective, widely used and is recommended by the WHO, etc.

- Chik vaccine (IXCHIK) is authorized in the US and QDENGA is authorized but EMA. So, other vaccines are available (though admittedly not widely available/deployed). 

Lines 66/67: PPT prevent bites, often by preventing contact, but not always. For example, it is not unusual for mosquitoes to contact (if only briefly) DEET protected skin. 

Lines 176 to 189. Perhaps move Tables 1 & 2 to supplemental materials? 

Line 203 to 235. CEA analysis.

- Was discounting rate (e.g., 3%?) applied to future costs/benefits? 

- Is the analyses perspective societal and/or health care system (vs individual)? 

- What was the CEA threshold, cost neutrality, i.e. no cost or better? 

Lines 367-384. 

- Was the assumption a closed population w/o immigartion/emigration, i.e. simulated “persons” were present for the entirety of the outbreak event? If yes, indicate such in text. Also, perhaps tough on its implication including, for example, that estimates would not be apt for persons who were not present for the entirety of an event such as travellers. 

- Use terminology more representative of what was done and that makes clear that estimates were generated with a model and are prone to limitations thereof. Perhaps along the lines of "...in our model simulations, a repellent with 100% initial efficacy after application and a half-life of 7 hours (as found for 372 dUDL with our survival analysis) reduces the estimated burden of ZKV infection by 30% (outbreak R0=2.2)...". 

- Highlight the asymmetry of the benefits function, i.e. the estimated population benefit:population coverage function increases in a positive fashion? An important upshot is that repellent coverage must be high to achieve substantial impact under the scenario(s) evaluated with the model [sidebar: I do not believe high levels of population coverage are achievable with PPT such as repellents, something that is generally borne out in the literature]. 

Lines 404-408. 

- Use terminology more representative of what was done and that makes clear that estimates were generated with a model and are prone to limitations thereof. Perhaps along the lines of "...in our CEA simulations, a repellent with 80% initial efficacy and twice daily application was estimated to avert total health care costs per person of...". 

Line 415 and Table 5. Consider adding text to indicate that the estimated cost per application to achieve the CEA threshold increases substantially (by > 20 x over the range of values considered) with decreasing population coverage? 

Lines 429 to 431. There is abundant literature to suggest low levels of compliance is often the case among travellers. 

Lines 500-511. Suggest a more fulsome discussion of whether current modalities, including dUDL, are likely to meet the target product profile for a PPT that would effectively reduce the burden of infection during an outbreak, i.e. high population coverage AND high efficacy.

PLOS authors have the option to publish the peer review history of their article (what does this mean?). If published, this will include your full peer review and any attached files.

Reviewer #1: Yes: Lisa Marie Rigby

Reviewer #2: No

Figure Files:

Data Requirements:

Reproducibility:

References

---

## [Editor Report · Decision Letter 1]

10 Oct 2024

Dear Dr Jones,

We are pleased to inform you that your manuscript 'The effectiveness of putative wearable repellent technologies to protect against mosquito biting and Aedes-borne diseases, and their economic impact.' has been provisionally accepted for publication in PLOS Neglected Tropical Diseases.

Best regards,

Nigel Beebe, PhD

Section Editor

Nigel Beebe

Section Editor

Thanks for the clear responses to referees comments - could you please also do a final check on the references as I noticed some species names not italicized - i.e. Ixodes sp.

---

## [Editor Report · Acceptance letter]

9 Dec 2024

Dear Dr Jones,

We are delighted to inform you that your manuscript, "The effectiveness of putative wearable repellent technologies to protect against mosquito biting and Aedes-borne diseases, and their economic impact.," has been formally accepted for publication in PLOS Neglected Tropical Diseases.

Best regards,

Shaden Kamhawi

co-Editor-in-Chief

Paul Brindley

co-Editor-in-Chief
